

# Semi-2DCAE: a semi-supervision 2D-CNN AutoEncoder model for feature representation and classification of encrypted traffic

Jun Cui[1,*], Longkun Bai[2,*], Guangxu Li[2], Zhigui Lin[2] and Penggao Zeng[1]

[1] Tiangong University, School of Life Sciences, Tianjin, China
[2] Tiangong University, School of Electronics and Information Engineering, Tianjin, China
[*] These authors contributed equally to this work.

Corresponding author
Zhigui Lin,
linzhigui@tiangong.edu.cn

## ABSTRACT

Traffic classification is essential in network-related areas such as network management, monitoring, and security. As the proportion of encrypted internet traffic rises, the accuracy of port-based and DPI-based traffic classification methods has declined. The methods based on machine learning and deep learning have effectively improved the accuracy of traffic classification, but they still suffer from inadequate extraction of traffic structure features and poor feature representativeness. This article proposes a model called Semi-supervision 2-Dimensional Convolution AutoEncoder (Semi-2DCAE). The model extracts the spatial structure features in the original network traffic by 2-dimensional convolution neural network (2D-CNN) and uses the autoencoder structure to downscale the data so that different traffic features are represented as spectral lines in different intervals of a one-dimensional standard coordinate system, which we call FlowSpectrum. In this article, the PRuLe activation function is added to the model to ensure the stability of the training process. We use the ISCX-VPN2016 dataset to test the classification effect of FlowSpectrum model. The experimental results show that the proposed model can characterize the encrypted traffic features in a one-dimensional coordinate system and classify Non-VPN encrypted traffic with an accuracy of up to 99.2%, which is about 7% better than the state-of-the-art solution, and VPN encrypted traffic with an accuracy of 98.3%, which is about 2% better than the state-of-the-art solution.

## INTRODUCTION

Traffic classification, as one of the fundamental problems in computer networks, is an important part of network traffic analysis. Traffic classification is mainly used to associate traffic into a specific class based on different requirements such as quality-of-service (QoS), routing improvement, and billing systems (*Azab et al., 2022*). So far, various traffic classification methods have been proposed. We can roughly classify these methods into

four types, including port-based methods, payload-based methods, machine learning-based (ML-based) methods, and deep learning-based (DL-based) methods.

Initially, traffic classification was performed using the port-based approach (*Moore & Papagiannaki, 2005*). Typically registered with IANA1 to represent well-known services (*Azab et al., 2022*), such as the standard SSH port of 22; Telnet's default port of 23, *etc*. This method is to extract the port number from the transport layer and determine the type of traffic based on different port information. Although the port-based traffic classification method is simple and fast, using known ports also vulnerable to attacks. Therefore, more and more network communications are beginning to use methods such as port masquerading, port randomization, and tunneling techniques to ensure their security, which makes port-based traffic methods no longer applicable. The payload-based traffic classification method is called deep packet inspection (DPI) (*Khalife, Hajjar & Díaz-Verdejo, 2017*; *Deri et al., 2014*). DPI technology classifies network traffic by analyzing the content of the payload portion of a data packet, which is more reliable than port-based traffic methods of encrypted traffic (*Bujlow, Carela-Español & Barlet-Ros, 2015*; *Lotfollahi et al., 2017*). In particular, many modern services use encryption technologies such as SSL to encrypt communications to protect user privacy, which poses a significant challenge to the application of DPI.

In recent years, researchers in the subject of traffic classification have given machine learning (ML) technology a lot of attention. In the ML method, statistical characteristics or time series characteristics (such as the number of data packets, average packet rate, maximum/minimum data packets, *etc*.) are obtained by analyzing the statistical data of the traffic. After acquiring the data features, ML models including K-nearest neighbors (KNN) (*Cover & Hart, 1967*), decision trees (DT) (*Quinlan, 1986*), and support vector machines (SVM) (*Cortes & Vapnik, 1995*) are used as classifiers for network traffic classification. Although the ML-based traffic classification method does not involve privacy security issues, there are still two shortcomings. First, large-scale traffic classification makes ML methods highly complex in both time and space. Secondly, data features are selected by researchers based on experience. These features not only require a large number of artificial resources but also cannot be guaranteed to reflect all characteristics of network traffic, which is time-consuming and prone to errors (*Sheikh & Peng, 2022*).

With the development of ML technology, a new generation of technology solutions–deep learning (DL) methods, has emerged and gradually matured. Currently, DL has achieved great success in applications such as computer vision (*Simonyan & Zisserman, 2014*; *He et al., 2016*) and natural language processing (*Peng et al., 2018*; *Yao, Mao & Luo, 2019*). Many researchers have begun to apply it to traffic classification, and have achieved good results (*Rezaei & Liu, 2019*; *Wang et al., 2019*). Mature DL models include convolutional neural networks (CNN) (*Zeng et al., 2019*; *Wang et al., 2017a*), recurrent neural networks (RNN) (*Lopez-Martin et al., 2017*; *Yao et al., 2019*), AutoEncoder (AE) (*Hinton & Salakhutdinov, 2006*; *Bourlard & Kamp, 1988*), *etc*. The DL model automatically learns differentiated features from the original network flow, which saves a lot of artificial resources and reduces the complexity of data processing. At the same time, when classifying large-scale network traffic, the DL model reduces the temporal and spatial complexity.

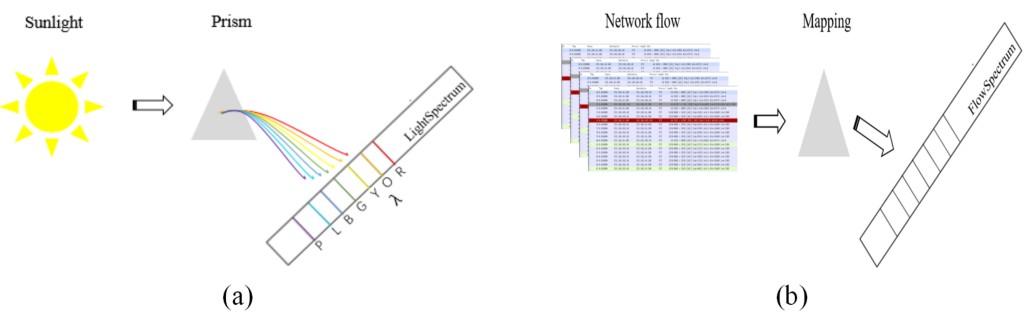

(a)                                           (b)

**Figure 1** **A simple comparison between light spectrum (A) and FlowSpectrum (B).** In (A) "R", "O", "Y", "G", "B", "I", and "P" represent "Red," "Orange", "Yellow", "Green", "Blue", "Indigo", and "Purple", respectively.

Although DL solves many shortcomings of ML in network traffic classification, there are also new problems. First, DL is an end-to-end learning strategy, which makes the neural network model a "black box" and poses a challenge for practical applications (*Xie, Li & Jiang, 2021*). Moreover, DL models rarely represent traffic features, which means that DL models are difficult to provide a basis for output results, which greatly hinders the application of DL in practical environments.

The concept of FlowSpectrum was first proposed in *Yang et al. (2022)*. Simply put, as shown in Fig. 1A, Sunlight can be separated into different colors by a prism because different colors of light have different wavelengths (λ). This creates a spectrum of light called the light spectrum. Similarly, as shown in Fig. 1B, network flows are mapped into a spectral line in a one-dimensional coordinate system through some kind of mapping, which we call the FlowSpectrum. FlowSpectrum is a new scheme for characterizing network flow (*Guo et al., 2022*). Its core is to construct a representation space of network behavior to represent and describe the behavior of network flow. In *Yang et al. (2022)*, researchers designed a deep autoencoder model to extract, decompose, and reduce the dimensionality of statistical features from the NSL-KDD dataset. Subsequently, the obtained features were represented as spectra in a one-dimensional coordinate system, with different types of flow features characterized within different spectral intervals. Finally, researchers use FlowSpectrum to detect and classify network traffic types. The FlowSpectrum provides a new approach for network flow analysis, which not only overcomes problems such as artificial feature selection and high space–time complexity in ML but also takes into account the problem of flow network characterization. However, the existing FlowSpectrum model still has some problems, such as poor generalization ability, insufficient extraction of spatial structure features of network flows, and low accuracy of network flow detection and classification.

In this article, we propose a Semi-2DCAE model for generating FlowSpectrum of encrypted flows and classifying encrypted flows. Deviating from the methodology proposed in *Yang et al. (2022)*, our approach aims to capture more robust spatial structural features of network flows by preserving the original byte sequences and transforming them into two-dimensional images. To achieve this, we employ a 2D-CNN architecture within our autoencoder model, enabling the extraction of salient features from the resulting

two-dimensional representations. This approach effectively addresses the previous limitations of FlowSpectrum models in extracting intrinsic structural characteristics of network flows. Empirical evaluation validates the effectiveness of our model in accurately classifying encrypted flows, outperforming previous FlowSpectrum models in this domain. Furthermore, we enhance the model's stability during the training process by incorporating the PRule activation function. Specifically, our scheme is divided into two stages: the training stage and the classification stage. In the training stage, firstly, we adopt the AutoEncoder (AE) model architecture (*Hinton & Salakhutdinov, 2006*), which makes our model capable of data dimensionality reduction. Then, in the AE structure, we add a 2-dimensional convolutional neural network (2D-CNN) layer, and then we process the original network packet into an image and save the turner IDX file (the original data is processed in session-level form) so that can then extract the original spatial structure information features. Next, we appropriately designed a semi-supervised learning strategy for deep learning (*Glennan, Leckie & Erfani, 2016*) to make the generated FlowSpectrum effective for encrypted traffic classification. Finally, we use the softmax function to classify the different spectral lines to form the FlowSpectrum. In the classification stage, we use a Bayesian optimal classifier (*Moore & Zuev, 2005*) to classify the test set based on the known FlowSpectrum. We can simply divide the task model into three components: encoder, mapper, and decoder. The encoder encodes the input data, and we use it to enrich the representation of the session-level input. Then, the mapper maps the encoded features into a spectral map to generate the FlowSpectrum. Next, we use the decoder for error reconstruction. Finally, using the generated FlowSpectrum as a criterion, we classify the features of the test data by a Bayesian optimal classifier to obtain the classification results.

To design Semi-2DCAE, there are two main technical challenges:

• Extraction of spatial structural features. Preserving the privacy of encrypted traffic prevents us from directly extracting information from the encrypted data. To address the difficulty of extracting spatial structural features of network flows, we incorporate a 2D-CNN within autoencoder model. To enable the application of the 2D-CNN network, we transform the original byte sequences of network flows into two-dimensional images.

• Stability of the FlowSpectrum model. Network flow data exhibits high complexity and differs from pixel values in images. The data values of network flows span the entire real number set, which can lead to instability during the model training process. To tackle this issue, we experimentally select the PReLU activation function, which is better suited for improving the stability of the model during the training process.

The contributions of this work can be summarized as follows:

• We propose a Semi-2DCAE model for generating a spectrum of encrypted flows and analyzing their spectral features. The Semi-2DCAE model utilizes the spatial feature extraction ability of 2D-CNN to form a semi-supervised autoencoder network. We use the Semi-2DCAE model to characterize the features of encrypted flows in a one-dimensional coordinate system for the first time, and it can be clearly seen that different encrypted flows exist in different interval ranges in the coordinate system.

• Selection of activation function for the model. Our experimental results have demonstrated that using the Prule activation function during the design process of deep

learning models for flow classification can result in a more stable training process. The Prule activation function retains values in the negative range, thereby avoiding unstable training caused by data loss.

- We use encrypted FlowSpectrum for encrypted flow classification to improve the accuracy of encrypted flow classification. In this article, we use the ISCX-VPN2016 dataset to obtain the FlowSpectrum after first training the Semi-2DCAE model using the training set, and then classifying the test set. The experimental results show that we improve the accuracy of classifying non-VPN encrypted flows and VPN encrypted flows by about 7% and about 2%, respectively, compared with the state-of-the-art methods.

The rest of this article is organized as follows. In 'Introduction', we introduce several common technical solutions for traffic classification and the research status related to FlowSpectrum. In 'Related work', we introduce the data types, the mechanism for flow classification using the FlowSpectrum, and the framework of the FlowSpectrum model. In 'Methodology', We have introduced the dataset used, experimental benchmarks, and model parameter settings. In 'Experiments', we conduct multiple experiments and compare the experimental results. Finally, we conclude this article and provided directions for future work in 'Results and discussion'.

## RELATED WORK

In the past decade, various methods have been proposed to address the problem of traffic classification. In this section, we will provide a detailed introduction to these methods.

### DPI-based methods

As mentioned earlier, port-based traffic classification is vulnerable to attacks, prompting the introduction of DPI technology. DPI has become one of the mature traffic classification technologies in recent years, and there are many traffic classification solutions based on DPI, including OpenDPI, nDPI, and Libprotoident. *Hubballi & Swarnkar (2018)* proposed a bit-level DPI technique called BitCoding for generating signatures. BitCoding identifies protocol types by using a small number of initial bits and using invariant bits as signatures. At the same time, to prevent conflicts and cross-signature matching caused by the increase in the number of signatures, the researchers use the Hamming distance variant to detect signature similarity and suggest increasing the signature length of some protocols to avoid overlap. The researchers conducted thorough experiments using three diverse datasets, confirming BitCoding's robust detection performance across various protocol types. *Alcock & Nelson (2012)* developed a DPI library called Libprotoident, which is suitable for application layer protocol identification of network flows. Libprotoident significantly saves disk space during flow processing primarily because it only utilizes the first four bytes of payload in each direction, packet size, and port number. Additionally, this approach reduces privacy concerns to some extent when dealing with encrypted flows. *Hubballi & Khandait (2022)* developed a DPI-based KeyClass traffic classifier that can classify network flows in a single scan payload using keyword-based signatures. KeyClass rapidly identifies potential applications by scanning a small number of initial bytes while continuously optimizing the number of character comparisons, which greatly improves classification

performance. The researchers tested the classification performance of KeyClass using two large datasets, with an average classification accuracy of approximately 98%.

Although DPI is a powerful traffic classification technology, it still has many problems. For example, DPI has a high computational complexity, potential privacy concerns during payload analysis, and the inability to identify encrypted traffic. This limits the application of DPI technology.

## Machine learning-based methods

In the last decade, there has been a significant amount of research conducted on the utilization of ML for traffic classification (*Shafiq, Yu & Wang, 2018*). ML-based methods differ from DPI in that they classify encrypted traffic by obtaining statistical features of data packets. When using ML for traffic classification, researchers first need to obtain features according to specific requirements (such as citation type/traffic type/protocol type). Then, flow features are input into algorithm models to obtain classification results. *Draper-Gil et al. (2016)* investigated the effectiveness of flow-based time-correlated features (*e.g.*, stream duration, bytes of streams a second, time before and after arrival, *etc*.) in classifying VPN flows. The researchers used two machine learning algorithms, DT and KNN, to test the accuracy of traffic classification based on time-correlated features. The experimental results showed an accuracy of over 80% for classifying VPN traffic. *Yamansavascilar et al. (2017)* evaluated the performance of four learning algorithms such as J48, random forest, K-NN, and Bayes Net for application identification and network traffic classification using the UNB-ISCX dataset and their internal dataset. The researchers used 111 features from the ISCX dataset in their experiments and eventually reduced the number of features in the feature set to 12 features. The experimental results show that the K-NN algorithm was able to achieve 93.94% accuracy on the ISCX dataset. *Wang (2015)* employed three ML algorithms, namely artificial neural network, DT, and SVM, for classifying network traffic and conducted a comparative analysis. Firstly, the researchers produced datasets by capturing online network traffic from seven different applications such as DNS, FTP, TELNET, P2P, WWW, IM, and MAIL. Then, they used the NetMate tool to extract the features of the captured packets. Finally, the extracted packet features were processed by three ML algorithms to classify the network traffics. The results of this experiment showed that the DT algorithm achieved a classification accuracy of 97.57%.

Although ML algorithms can address some issues encountered in DPI, including data privacy protection, they also introduce new challenges. Firstly, the classification performance of ML algorithms for network flows heavily relies on the crucial step of feature selection. Correctly identifying and extracting features relevant to traffic classification is paramount for algorithm performance. However, determining the optimal feature set presents a challenge, requiring domain expertise and experimental adjustments, which increases the resource overhead for traffic classification. Secondly, the high computational complexity of machine learning algorithms results in significant delays during the training and testing phases, impacting their practical application in traffic classification. Lastly, traditional ML algorithms (*e.g.*, KNN, DT, SVM, *etc*. mentioned above) are of the supervised

learning type and require a large number of traffic labels to be added manually, which likewise increases the difficulty of applying ML to traffic classification.

## Deep learning-based methods

The difference between DL-based and ML-based methods is that DL-based methods do not require manual feature selection. They are end-to-end automatic learning processes that eliminate the feature selection stage, making them more convenient and efficient traffic classification solutions. *Wang et al. (2017b)* proposed a method for classifying malware traffic using traffic data as images based on 2D-CNN networks. Instead of designing features manually, the method directly feeds the raw data flows into the model as the input data of the classifier for classification. The experimental results show an average classification accuracy of 99.41%, demonstrating the effectiveness of the method in malicious traffic classification. *Wang et al., (2017a)* proposed an end-to-end encrypted traffic classification method using one-dimensional convolutional neural networks (1D-CNN). This approach integrates feature extraction, feature selection, and classifier into a unified framework, enabling the automatic learning of nonlinear relationships between the raw input and its corresponding output. This marks the first application of an end-to-end deep learning model in the field of encrypted traffic classification. The team processed the original network flows at both the flow level and session level. Experimental results demonstrate that the end-to-end 1D-CNN model effectively enhances the accuracy of encrypted traffic classification, with the session-level classification outperforming the flow-level classification in terms of accuracy. *Xie, Li & Jiang (2021)* proposed a self-attention model called SAM for traffic classification. The SAM model comprises four components: an embedding layer, a self-attention layer, a 1D-CNN network layer, and a classifier. The research team conducted experiments using three different types of datasets, including the WIDE dataset for protocol classification, the UNIBS dataset for application classification, and the ISCX dataset for flow type classification. Among these, the experimental results revealed a classification accuracy of 90.3% for the ISCX dataset. Furthermore, the research team employed the self-attention mechanism to allocate attention weights to network flow features used for classification, providing insights into the discriminative basis of the DL model. *Lopez-Martin et al. (2017)* proposed a combined CNN and RNN (specifically LSTM) model for Internet of Things (IoT) traffic classification. The research team utilized the RedIRIS dataset and extracted the first 20 packet information from each network flow with the same five-tuple (source port, destination port, source IP, destination IP, and protocol). Instead of relying on payload data, the team extracted data based on advanced headers, which reduced model training latency and decreased the volume of data features. Experimental results demonstrate that the model effectively classifies IoT traffic without requiring any feature engineering during the process. *Höchst et al. (2017)* proposed an unsupervised traffic classification method that leverages statistical features of traffic and a dimensionality reduction scheme based on AE. The research team employed a time interval-based feature vector construction method and a semi-automatic clustering labeling approach. The evaluation was conducted on approximately four months of real data captured from around 25 mobile devices. Experimental results demonstrate that the proposed method successfully detects seven

different types of network flows, achieving an average accuracy of 80% and an average recall rate of 75%. *Li et al. (2018)* proposed an improved Stacked AutoEncoder (SAE) approach to learn complex relationships among multiple source network flows for classification. By stacking multiple basic Bayesian AutoEncoders, the research team achieved network flows classification. The proposed method was evaluated on synthetic datasets based on the MAWI and DARPA99 datasets. The experimental results demonstrated a classification accuracy of 83.2%.

Due to the ability of DL models in traffic classification to automatically learn feature representations from flow data without relying on handcrafted features, DL models have simplified the application of traffic classification. Furthermore, DL models can extract more abstract and useful features from raw network flows, better capturing the intrinsic structural characteristics of the flows compared to ML models. Additionally, unsupervised traffic classification models based on DL (*Höchst et al., 2017*; *Li et al., 2018*) reduce the dependency on labels, leading to substantial savings in labor and resources compared to supervised traffic classification models based on DL and ML. However, there are some challenges in this regard. Firstly, in the process of traffic classification, unsupervised classification models such as AE and SAE have lower accuracy compared to supervised classification models. Secondly, flow representation, as a crucial task for simplifying network analysis, requires the characterization of network flows in a simplified domain. Unfortunately, we have observed that both ML-based and DL-based traffic classification approaches have paid little attention to this critical task of flow representation.

## FlowSpecturm-based methods

FlowSpectrum, as introduced in the first section, is a novel approach to traffic analysis that provides a specific representation method for discernible features of network traffic in the cyberspace. The core of FlowSpectrum theory lies in simplifying the analysis of sparse high-dimensional network traffic features through a simplified domain representation. This simplification enhances the performance of network flow classification, threat attack detection, and interception of anomalous traffic, thereby enabling service providers to maintain high quality of service (QoS) and service availability. *Yang et al. (2022)* proposed the FlowSpectrum theory for the first time and designed a neural network architecture based on semi-supervised AutoEncoder for network flow data decomposition and dimensionality reduction. Specifically, the research team utilized the semi-supervised AutoEncoder model to decompose and reduce the features of the NSL-KDD dataset, mapping them onto a one-dimensional standard coordinate system to form feature spectra. Different types of flow features are reflected in different intervals within the coordinate system. The team tested the feature representation and intrusion detection capability of FlowSpectrum using the NSL-KDD intrusion detection dataset, achieving a preliminary correspondence between network behavior and spectral domain, as well as intrusion detection capability. In *Guo et al. (2022)*, extended the FlowSpectrum theory by proposing fundamental methods to map network flow data from the original flow space to the FlowSpectrum space. This included supplementary mathematical theories of FlowSpectrum and the basic principles of constructing FlowSpectrum models. Additionally, the research team

conducted experiments using the UNSW-NB15 dataset based on the FlowSpectrum theory. For the first time, they mapped the features of nine types of attack traffic from the UNSW-NB15 dataset onto a two-dimensional coordinate system and classified the nine types of flows, achieving a classification accuracy of 67.72%. It is evident that designing a stable, highly generalized, and effective FlowSpectrum mapping model crucially relies on the model's ability to extract spatiotemporal features of network flows and perform dimensionality reduction. *Fu et al. (2022)* designed an online attack detection model called Whisper, which uses the discrete Fourier transform (DFT) algorithm to successfully map network flows to an effective frequency domain space. Experimental results show that this method has an accuracy rate of over 95.00% in detecting threat attacks. *Bouzida et al. (2004)* proposed a PCA-based (*Tipping & Bishop, 1999*) network data dimensionality reduction method and used it for threat attack detection. *Hu, Gu & Wei (2021)* designed a CLD-Net model that reduces high-dimensional features learned by neural networks to eight dimensions to distinguish eight different types of network encryption traffic through feature reduction. *Shi et al. (2018)* proposed a feature optimization method based on DL and feature selection (FS) technology. Based on real traffic trajectory data, experimental results show that the proposed method can not only effectively reduce the dimensionality of the feature space but also overcome the negative effects of multi-class imbalance and concept drift on ML technology. *He et al. (2022)* proposed a Boruta-ET model based on Boruta and extreme tree (ET) algorithms, which uses Boruta-based algorithms to reduce network flow features. The goal of Boruta dimensionality reduction is to extract all features related to the dependent variables with global dimensions and find the best subset of features containing the most information. Finally, the optimal feature subset is used as an input parameter for the ET algorithm model for training and testing. *Imran et al. (2012)* used linear discriminant analysis (LDA) and genetic algorithms (GA) for feature transformation and optimal subset selection.

In general, there are some limitations in the current FlowSpectrum research. Firstly, FlowSpectrum models exhibit weak generalization capability and low detection/classification accuracy. For example, the model proposed in *Yang et al. (2022)* is not suitable for encrypted flows, and the classification accuracy of the nine types of attack traffic in *Guo et al. (2022)* is only 67.72%. Secondly, the research has overlooked the structural characteristics of network flow space. In complex network environments, the spatial structural features of network flows often carry crucial information. However, existing FlowSpectrum models cannot extract the original structural features of network flow space.

To address the limitations of different approaches in network flow classification and improve and expand existing FlowSpectrum techniques, this study proposes a Semi-2DCAE model. The model is designed to generate FlowSpectrum for encrypted flows and utilize these FlowSpectrum for flow classification. Our approach has two main advantages. First, we process the original network traffic into two-dimensional graphs at the session level and extract their spatial features using a 2D-CNN network's spatial feature extraction capability. Overall, we generate encrypted traffic spectra through the Semi-2DCAE model and represent the spatial features of Non-VPN encrypted traffic and VPN encrypted traffic

in a one-dimensional coordinate system, which provides a basis for the classification results. Second, our model utilizes the PReLU activation function, making it more suitable for processing network flow data and improving its stability. Third, we classify the network traffic in the test set using these spectra and achieve higher accuracy than state-of-the-art methods for classifying encrypted traffic.

## METHODOLOGY

As mentioned, the Semi-2DCAE model aims to map network traffic into FlowSpectrum and classify encrypted traffic. To achieve this, we propose utilizing a 2-dimensional convolutional autoencoder network to reduce the dimensionality of the network traffic and extract meaningful information. In this section, we will cover the choice of input data type in 'Input type', then explain the FlowSpectrum mechanism in 'FlowSpectrum Mechanism', and finally introduce the entire framework structure of Semi-2DCAE in 'Frame Design'.

### Input type

In the study of network traffic analysis, the most common representations of traffic are divided into the following three categories:

    (i) Packet-level based;

    (ii) Flow-level based;

    (iii) Session-level based.

    As shown in Fig. 2, a packet-level representation based on *Xie, Li & Jiang (2021)* primarily extracts information for each packet in a network flow, such as bytes from the data link, IP, TCP/UDP, and application layers. However, this approach ignores the directional information between flows and hosts. As shown in Fig. 3A, flow-level representation (*Fu et al., 2022*) mainly involves counting packets between unidirectional source IP, source port, destination IP, destination port, and transport protocol. Figure 3B shows session-level representation (*Wang et al., 2017a*), which is similar to flow-level representation, but with the difference that session-level representation is a bidirectional representation of a flow, ensuring the flow direction. Session-level representation enhances the correlation between original packets, greatly increasing the richness of extracted features, and improving the accuracy of traffic classification. Therefore, in this study, we adopt the session-level representation to represent network flows and describe our data handling approach in the following section.

    During data preprocessing, we represent the raw network flows as sessions and extract features from each packet. To enable the use of a CNN as the first layer, it is necessary to ensure that each input data has the same size. We adopt the suggestion from *Wang et al. (2017b)* and extract the first 784 bytes of each packet, converting the data into grayscale images of size 28 × 28 for final representation. As shown in Fig. 4, the original pcap files are initially segmented into discrete session files. Subsequently, incomplete or duplicate packets are removed from the cleaned files, and the resulting file is then cropped to 784

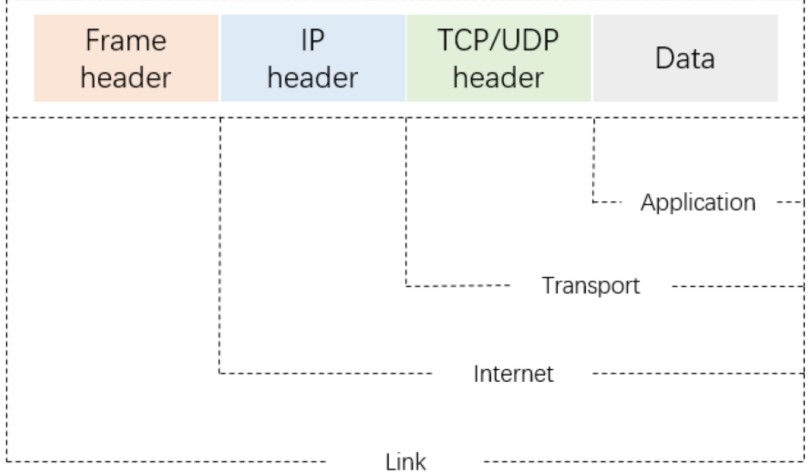

**Figure 2  The four-layer structure model for IP packets.**

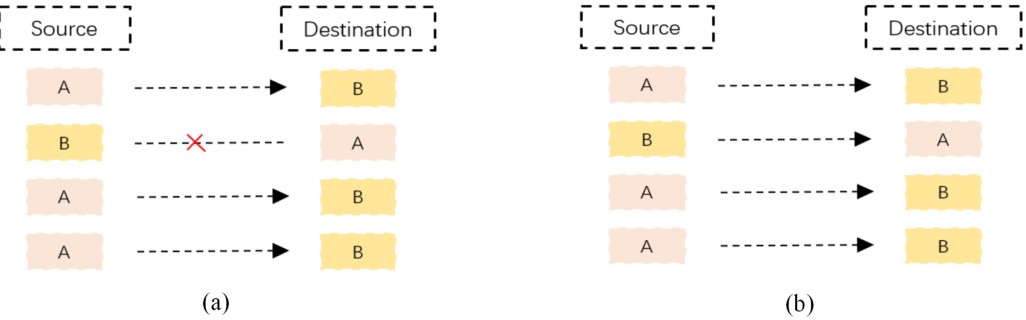

**Figure 3  Schematic diagram of network flow and network session.** (A) Shows the diagram of a flow. Where "A" and "B" indicate two IP addresses and "×" means no access; (B) a diagram of a session.

bytes. If a file is smaller than 784 bytes, it is filled with $0 \times 00$. Finally, the files with a standardized size of 784 bytes are saved in IDX format.

## FlowSpectrum mechanism

As described in 'Introduction' and 'Related work', the FlowSpectrum is a scheme for characterizing network flow features. Generating FlowSpectrum using a FlowSpectrum model and classifying network flows based on them is the application of this article.

### Representation of FlowSpectrum

We first define the set of network flow features as the vector $\vec{F} = \{f_1, f_2, \ldots, f_m\}$, here $f_i$ is the eigenvalue, $i \in [1, m]$. For each instance flow, we define this as $\vec{x} = \{x_1, x_2, \ldots, x_m\}$, here $x_i$ is the instance flow point value, corresponds to the value of $f_i$ in $\vec{F}$. We define a set of instance flows as $\mathcal{X} = \{\vec{x}_1, \vec{x}_2, \ldots, \vec{x}_n\}$, here $\vec{x}_w \subseteq \mathcal{X} \subseteq \mathbb{R}^m$, $\mathcal{X}$ is an object in cyberspace, $\mathbb{R}$ is a real number space object, $w \in [1, n]$. Also, we define the instance output set as $\mathcal{L} = \{t_1, t_2, \ldots, t_k\}$, here each $t$ indicates an output type. We create a set $\mathcal{D} = \left\{ (\vec{x}_1, \vec{L}_1), (\vec{x}_2, \vec{L}_2), \ldots, (\vec{x}_m, \vec{L}_m) \right\}$,

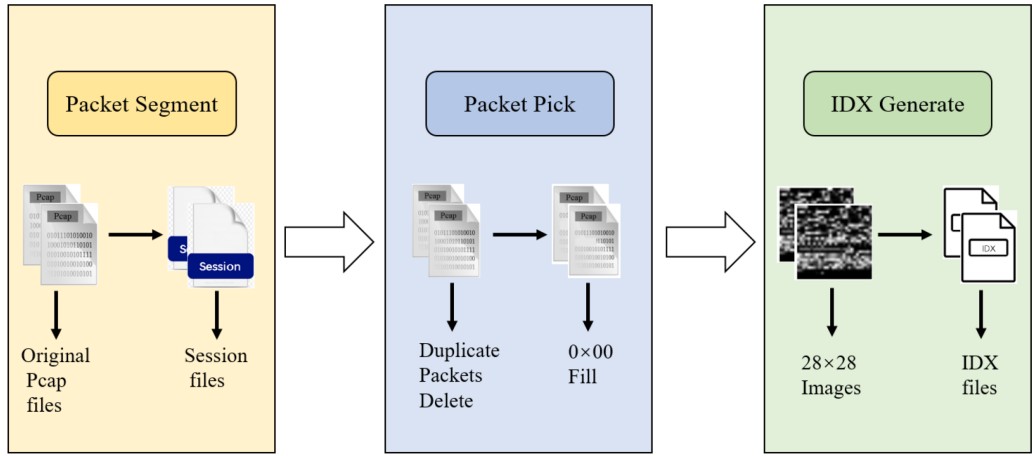

**Figure 4   Input data pre-processing diagram.**

which is the corresponding set of input instances $\vec{x}$ and output instances $\vec{L}$, $\vec{L_w} \in \mathcal{L}$. We denote by $\mu_t(\vec{x})$ the probability that the output type of $\vec{x}$ is $t$. Expressed in the formula as follows:

$$\mu_t(\vec{x}) = P\{Y = t \,||\, X = \vec{x}\} \tag{1}$$

where $Y$ and $X$ are respectively a random variable in $\mathcal{L}$ and $\mathcal{X}$. Finally, we define an $\mathfrak{F}(\mathcal{X}_t)$ to denote the set of feature values of the network space flow mapping to the real number space (*i.e.*, $\mathcal{X} \rightarrow \mathbb{R}$):

$$\mathfrak{F}(\mathcal{X}_t) = \begin{cases} d(\vec{x_1}) : \mu_t(\vec{x_1}), \\ d(\vec{x_2}) : \mu_t(\vec{x_2}), \\ \cdots, \\ d(\vec{x_n}) : \mu_t(\vec{x_n}) \end{cases} \tag{2}$$

where $d(\vec{x_n})$ is the value in the real number space $\mathbb{R}$. We call $\mathfrak{F}(\mathcal{X}_t)$ the FlowSpectrum. As shown in Fig. 5, we provide a FlowSpectrum mechanism flowchart.

### FlowSpectrum applied to traffic classification

The process of generating FlowSpectrum belongs to the decomposition and representation of network flows. As described in Section 1 FlowSpectrum generation is part of the training phase, while flow classification belongs to the classification phase. In the classification phase, the classifier we use is the Bayesian optimal classifier. First, we create the test flows set $\mathcal{X}' = \{\vec{x_1}, \vec{x_2}, \ldots, \vec{x_n}\}$, where the probability value of instance $\vec{x_1}$ belonging to a certain category is set to $P(\vec{x_1} || \mathcal{X}')$. Then, the FlowSpectrum of the test flows is as follows:

$$\mathfrak{F}(\mathcal{X}') = \begin{cases} d(\vec{x_1}) : P(\vec{x_1} || \mathcal{X}'), \\ d(\vec{x_2}) : P(\vec{x_2} || \mathcal{X}'), \\ \cdots, \\ d(\vec{x_n}) : P(\vec{x_2} || \mathcal{X}') \end{cases} \tag{3}$$

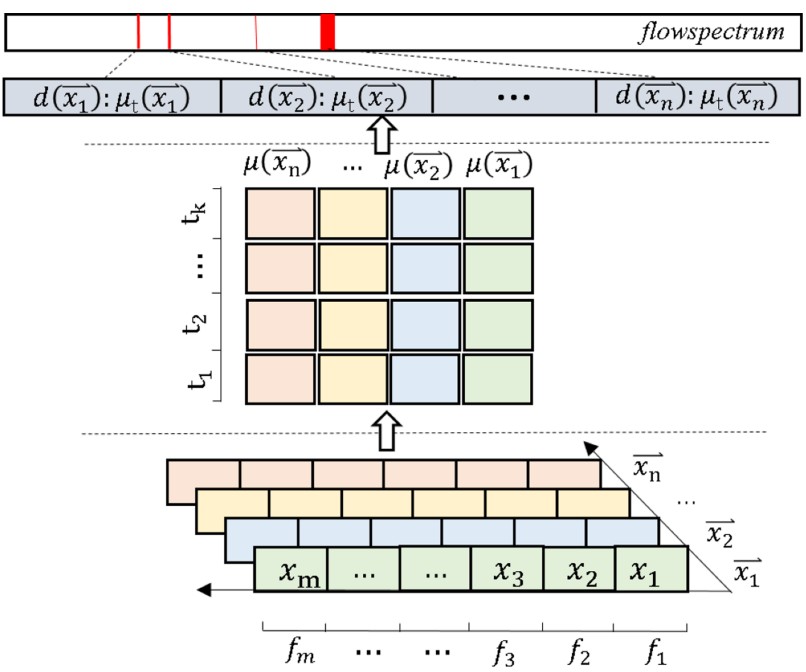

**Figure 5** Input data pre-processing diagram.

And then, the similarity probability of the FlowSpectrum is found by clicking the formula as follows:

$$\mathfrak{F}(\mathcal{X}') \cdot \mathfrak{F}(\mathcal{X}_t) = P\{Y = t \mid\mid X \in \mathcal{X}'\} \tag{4}$$

where $P\{Y = t \mid\mid X \in \mathcal{X}'\}$ is the probability that $\mathcal{X}'$ belongs to $\mathcal{X}_t$. Finally, the final classification result is found by finding the maximum likelihood estimate $\hat{y}$. This is expressed as follows:

$$\hat{y} = \arg\ \max P\left\{Y = t \mid\mid \mid X \in \mathcal{X}'\right\} = \arg\ \max \mathfrak{F}\left(\mathcal{X}'\right) \cdot \mathfrak{F}(\mathcal{X}_t). \tag{5}$$

when there are uncharacterized spectral values in $\mathcal{X}'$, we need to calculate the minimum distance between the spectral lines to determine the type of unknown spectral lines. We use the calculation method of exponential decay proposed in *Yang et al. (2022)*, as follows:

$$\vec{x}' = \arg\ \min_{x' \in x_t} \left\| d(\vec{x}) - d\left(\vec{x}'\right) \right\| \tag{6}$$

$$\mu_t(\vec{x}) = \mu_t\left(\vec{x}'\right) e^{-\left\| d(\vec{x}) - d\left(\vec{x}'\right) \right\|} \tag{7}$$

where $\vec{x}'$ is the test instance flow.

### Frame design

As shown in Fig. 6, Semi-2DCAE mainly consists of encoder, mapper, and decoder, where the encoder is composed of the input layer, 2D-CNN, and MaxPooling2D; the mapper is

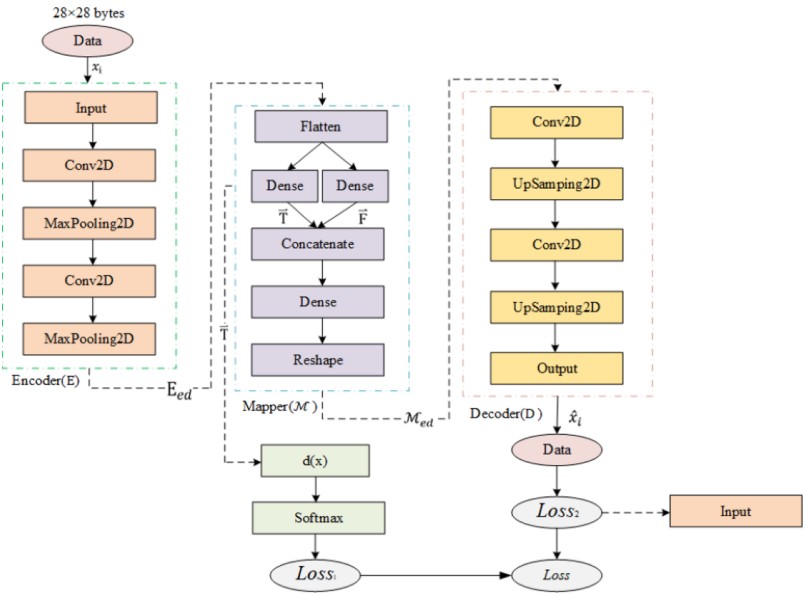

**Figure 6** Semi-2DCAE model structure.

composed of flatten layer, Dense layer, Concatenate layer, Reshape layer, and finally using softmax function for classification; the decoder is composed of 2D-CNN, UpSampling2D and output layer.

### Encoder

The encoder consists of an input layer, 2D convolutional layer, PRelu layer, and pooling layer, and its function is to extract features from the network flow. The model receives 28x28 byte flow images and trains 2D convolutional kernels in the convolutional layer. The convolutional layer is represented as follows:

$$H^j = f\left(b^j + \sum_i w^{ij} * x_i\right) \tag{8}$$

$H^j$ and $x_i$ are the jth output mapping and the ith input mapping respectively. $w^{ij}$ denotes convolution filter weights. "$*$" indicates convolution, $b^j$ is the deviation parameter of the jth mapping. $f$ denotes the activation function. For this convolutional layer, the PRelu activation function is used to prevent overfitting and to fully explore the correlations between features (we will provide a detailed introduction to activation functions in 'Comparison of feature representation of encrypted traffic'). We then applied a maxpooling layer to reduce redundant data and compress it. Pooling also increases the receptive field, making the pooled data features translation invariant. Overall, we represent the encoder as E, with the input network flow as $\vec{x}$ and $E_{ed}$ defined as the encoded network flow. Then, the encoder is represented in the following formula:

$$E_{ed} = E[H(\vec{x})]. \tag{9}$$

### Mapper

The mapper consists of a flattening layer, fully connected layer, reshaping layer, and Softmax classifier, and its function is to generate FlowSpectrum by representing the network flow features as spectral lines. After encoding, the high-dimensional data features become multidimensional vector features in the form $E_{ed}$. Then, the feature maps are smoothed and mapped to two real-valued numbers using two fully connected layers. We use the notation $\vec{T}$ to represent the feature that can be characterized and $\vec{F}$ to represent the feature that cannot be characterized, which correspond to the two real-valued numbers. And then, we input the representable feature $\vec{T}$ into the Softmax function to get the probability value that the flow belongs to a certain category, and take the category with the largest probability value as the final classification result. The formula of the Softmax function is shown below:

$$p_i = \frac{e^{T_i}}{\sum_{i=0}^{k} e^{T_i}} = \frac{e^{d(x_i)}}{\sum_{i=0}^{k} e^{d(x_i)}} \tag{10}$$

where $p_i$ is the probability that the input sample belongs to category i, $T_i \in \vec{T}$ is a real number in the range, i is the encrypted flows category index, and k is the total number of encrypted flows categories. Here, we have $d(x_i) = T_i$, i.e., our FlowSpectrum set $d(x) = \vec{T}$. After obtaining the classification results, we use the categorical cross-entropy function to calculate the $Loss_1$:

$$Loss_1 = -\sum \left\{ X_i^n \log_2 \left( p_i^n \right) \right\} \tag{11}$$

where n is the sample label index. In addition, we use the representable features $\vec{T}$ and non-representable features $\vec{F}$ as the input of the decoder after concatenation and shaping.

Overall, we denote the mapper as $\mathcal{M}$. By continuously training $Loss_1$ value decreases, we finally get the FlowSpectrum of representable features. Denote $\mathcal{M}_{ed}$ as the mapping output and $\oplus$ as the concatenated shaping operator. Then the mapper is represented by the following equation:

$$\mathcal{M}_{ed} = \vec{T} \oplus \vec{F} \tag{12}$$

### Decoder

Also known as the reconstructor, after the mapper outputs $\mathcal{M}_{ed}$, the decoder is used to reconstruct the original network flows to obtain the corresponding reconstructed value. We use the mean square error function to calculate the $Loss_2$:

$$Loss_2 = \frac{1}{n} \sum_{i=1}^{n} \| \hat{x}_i - x_i \|^2 \tag{13}$$

where $\hat{x}_i$ is the element value of the reconstructed network flows $\hat{x}$.

Overall, we denote the decoder as $\mathcal{D}$. Then in the decoder is represented by the following equation:

$$\hat{x} = \mathcal{D}(\mathcal{M}_{ed}) \tag{14}$$

The loss value of our whole framework design is set to *Loss*, then that Loss is the sum of the *Loss* weights of the mapper and decoder.

## EXPERIMENTS

In order to demonstrate the performance of the Semi-2DCAE FlowSpectrum model, we evaluated our scheme through comprehensive experiments. In 'Dataset', we first introduce the data set used in the experiment; In 'Scheme and indicators', we set the benchmark of the experiment and the evaluation index; Finally, we describe our model setting parameters in 'Parameter selection', and prove the advantages of using the PRule activation function.

### Dataset

In terms of encryption traffic classification, we use the original data packets of the ISCX-VPN2016 dataset (*Draper-Gil et al., 2016*), which contains Non-VPN encryption traffic and VPN encryption traffic. Each encryption type includes six types of traffic: chat, Email, File, P2P, Streaming, and VoIP. Note that we have deleted the packets without specific labels, so it is less than one described in *Draper-Gil et al. (2016)*. In addition, we also used the NSL-KDD data set(CSV file) (*Tavallaee et al., 2009*), which contains five types of traffic: normal, DOS, Probing, U2R, and R2L. The NSL-KDD data set is manually processed, and each type of traffic has labels, so it is significantly different from the ISCX-VPN2016 data set we use. Note that we only use the NSL-KDD data set to compare with the article (*Yang et al., 2022*). To illustrate that there are spatial structure features in the original network data, our model can capture this feature.

### Scheme and indicators

To demonstrate our FlowSpectrum characterisation capabilities and the effectiveness of cryptographic traffic classification, We have established three types of baselines:

(i) Machine learning-based: we choose the popular SVM (*Cortes & Vapnik, 1995*).

(ii) Deep learning-based: we choose the 1D-CNN model (*Wang et al., 2017a*) and CNN+RNN (*Yao et al., 2019*).

(iii) FlowSpectrum-based: we choose the Semi-AE model, which first proposed the FlowSpectrum theory and model (*Yang et al., 2022*), and we also build a new self-built FlowSpectrum model–Semi-1DCAE, which mainly uses 1D-CNN as a semi-supervised autoencoding network.

All these methods were run on a server with 8CPUs x Intel(R) Xeon(R) Platinum 8375C CPU @ 2.90 GHz and RTX 3090 Ti GPU. python version 3.9.0.

In order to evaluate the performance of different methods, we use four indicators: accuracy (ACC), Precision, Recall and F1 score(F1) as experimental evaluation. Each value is calculated as follows:

$$ACC = \frac{TP + TN}{TP + FP + FN + TN} \tag{15}$$

$$Precision = \frac{TP}{TP + FP} \tag{16}$$

$$Recall = \frac{TP}{TP+FN} \tag{17}$$

$$F1 = \frac{2Recall * Precision}{Precision+Recall} \tag{18}$$

Among them: TP, the number of traffic correctly allocated to a specific category; FP, the number of traffic incorrectly allocated to a specific category; FN, the number of traffic that belongs to a specific category but is allocated to other categories; TN, the number of traffic that not belongs to a specific and is allocated to other categories.

## Parameter selection
### Hyperparameters
In the Semi-2DCAE model, the size of convolution cores in the convolution layer is 3, the number of convolution cores used in the first and second convolution is 32 and 64, respectively, and the size of the maximum pooling layer is 2. It should be noted that we use L2 regularization in the encoder to prevent over-fitting, and the regularization rate is 0.01. Epoch is set to 300, Batch_size is 64, and the learning rate is 0.0005.

### Activate the function
Our model selects the random linear rectification activation function PRelu (*Kannari, Shariff & Biradar, 2021*) with leakage, whose expression is:

$$f(x) = \begin{cases} x & x > 0 \\ \lambda x & x \le 0 \end{cases}, \lambda \in \cup(L,u), L < u, u \in [0,1) \tag{19}$$

$x$ is the transmission data of neurons, $\lambda$ is the gradient function of PRelu when the input value is a negative value range, and $\cup(L,u)$ represents the continuous uniform distribution function probability model. $\lambda$ is a random variable drawn from $\cup(L,u)$. Through formula (19), we can find that PRelu can update the weight normally when the neuron output data is greater than zero, while the weight can also be updated slightly when the neuron output data falls within the negative range of zero. As shown in Figs. 7 and 8, during the model training phase (note that during the training phase, we divide the training dataset into training and validation sets in a 2:1 ratio.), we provided the training accuracy and training loss of the model using two different activation functions, PRelu and Relu. From Figs. 7A and 8A, it can be observed that when Relu is used as the activation function for the model, oscillations occur during the training process. This phenomenon arises due to the output of the activation function being zero, preventing the update of weights when data values fall within the negative range, coupled with the presence of a large learning rate. Paradoxically, reducing the learning rate hampers the model's ability to converge to the global minimum. However, from Figs. 7B and 8B, it is evident that when PRelu is employed as the activation function, our model exhibits greater stability during the training process, leading to an improvement in training accuracy. By choosing PRelu as the activation function, our model effectively mitigates the learning rate issue, ensuring convergence to the global minimum and preventing overfitting.

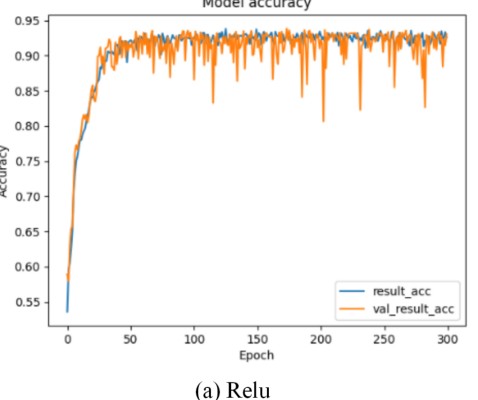
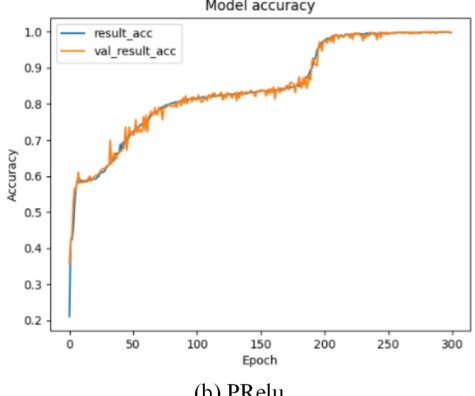

(a) Relu                                    (b) PRelu

**Figure 7** **Training accuracy of Semi-2DCAE model.** Among them, 'result_acc' represents the accuracy on the training set, and 'val_result_acc' represents the accuracy on the validation set.

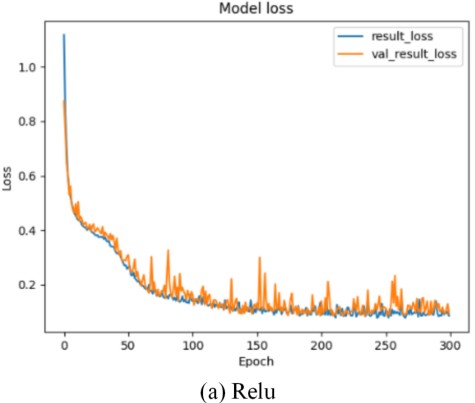
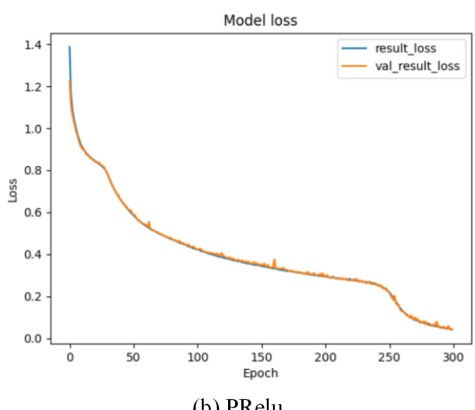

(a) Relu                                    (b) PRelu

**Figure 8** **Training loss of Semi-2DCAE model.** Among them, 'result_loss' represents the loss on the training set, and 'val_result_loss' represents the loss on the validation set.

## RESULTS AND DISCUSSION

To demonstrate the performance of Semi-2DCAE, we evaluate our scheme through comprehensive experiments. In this chapter, we give the experimental results and make a comparative analysis. First of all, in comparison of 'FlowSpectrum prototype data', we compared and analyzed the performance of our Semi-2DCAE model and the Semi-AE model in *Yang et al. (2022)* on the NSL-KDD dataset through experiments. In 'Comparison of feature representation of encrypted traffic', a comparative analysis was conducted between the proposed Semi-2DCAE model and two baseline models (Semi-AE and Semi-1DCAE) to assess their effectiveness in feature representation for encrypted traffic. In 'Comparison of encrypted traffic classification', the results of different models for encrypted traffic classification were presented, followed by a detailed comparative analysis of these results.

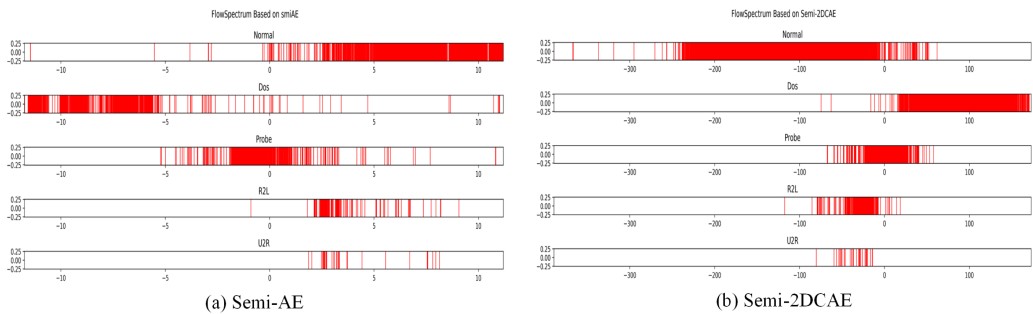

**Figure 9** FlowSpectrum of the statistical features of the NSL-KDD dataset based on the Semi-AE model and Semi-2DCAE model.

**Table 1** Comparison of recall rate of NSL-KDD data set.

| Type | Semi-AE | Semi-2DCAE |
| --- | --- | --- |
| Normal | 0.980 | 1.000 |
| Dos | 0.996 | 0.941 |
| Probe | 0.617 | 0.552 |
| Average | 0.864 | 0.831 |

## Comparison of FlowSpectrum prototype data

We performed classification on the NSL-KDD dataset using the Semi-2DCAE model and the Semi-AE model described in *Yang et al. (2022)*. Because the NSL-KDD data set is a statistical feature after manual filtering, it can be compared with the original network flow feature directly extracted in this article. As shown in Fig. 9, we obtained FlowSpectrums for the NSL-KDD dataset using both the Semi-2DCAE and Semi-AE models. From Fig. 9B, It can be observed that the FlowSpectrum lines generated by our Semi-2DCAE model exhibit overlaps (*e.g.*, Normal, DoS, and Probe lines). Additionally, as presented in Table 1, we compared the recall rates of the NSL-KDD dataset using the Semi-2DCAE and Semi-AE models (notably, the R2L and U2R data were excluded due to their minimal proportions, as specified in *Yang et al., 2022*). The average recall rates for the Semi-2DCAE and Semi-AE models were 83.1% and 86.4%, respectively. Overall, our Semi-2DCAE model demonstrates relatively weak representation capability for the manually selected statistical features in the NSL-KDD dataset. However, in terms of recall rates, our model is closely aligned with the flow-based Semi-AE model. Furthermore, in the subsequent experimental analysis of encrypted traffic classification, we anticipate entirely different results when extracting and classifying the raw data features of encrypted traffic. It should be pointed out that based on the source code and data provided in *Yang et al. (2022)*, we did not obtain a result with an average recall rate of 95.13% in *Yang et al. (2022)*. Although we attempted to modify the source code to improve the recall rate, we were unsuccessful.

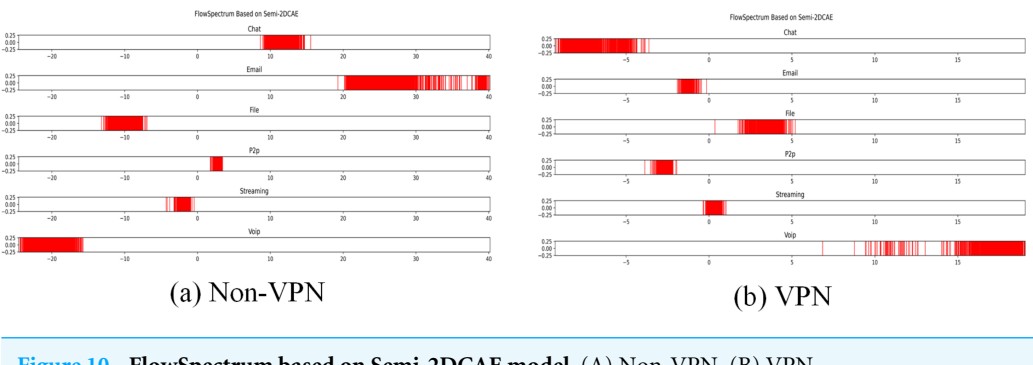

(a) Non-VPN                    (b) VPN

**Figure 10  FlowSpectrum based on Semi-2DCAE model.** (A) Non-VPN. (B) VPN.

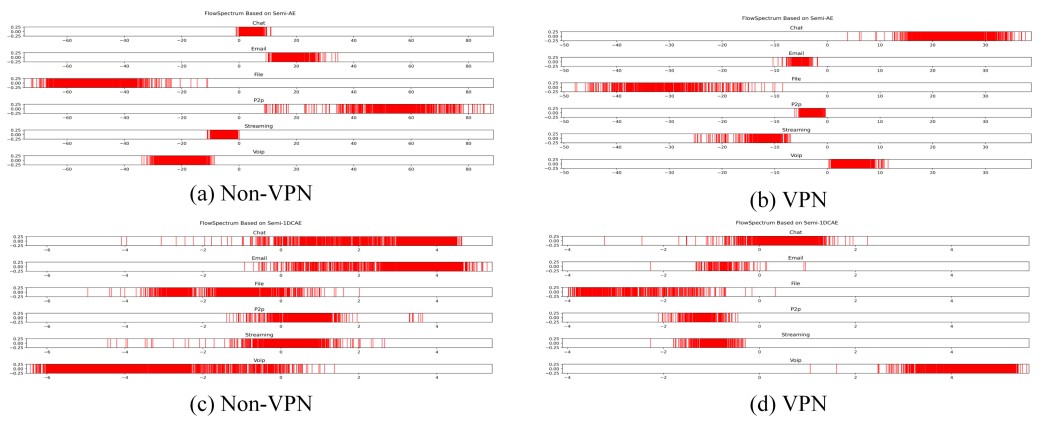

(a) Non-VPN                    (b) VPN

(c) Non-VPN                    (d) VPN

**Figure 11  FlowSpectrum based on Semi-AE model and Semi-1DCAE model.** (A) Non-VPN. (B) VPN.

## Comparison of feature representation of encrypted traffic

In the ISCX-VPN2016 dataset, there are two types of encrypted traffic: Non-VPN encryption and VPN encryption. Each encryption type further consists of six different traffic types, namely Chat, Email, File, P2P, Streaming, and VoIP. For each traffic type, we have generated FlowSpectrum, which represents the low-dimensional features of the network flows. Our objective is to analyze the characteristics of different types of network flows based on the FlowSpectrum. Figure 10 illustrates the range of feature representation spectra for Non-VPN encrypted flows and VPN encrypted flows generated by the Semi-2DCAE model proposed in this article. From Fig. 10, it can be observed that different types of traffic correspond to different intervals on the spectra. For instance, in the Non-VPN encrypted traffic, the FlowSpectrum intervals of chat flows are mostly distributed in the range of (8, 14), while the range of (20, 40) represents the concentration of email flows. In the case of VPN encrypted traffic, the FlowSpectrum of chat traffic mainly falls within the interval of (−5, −10), while the range of (0, −2) represents the distribution of email traffic. Figure 11 illustrates the range of feature representation spectra for Non-VPN encrypted flows and VPN encrypted flows generated using two benchmark models, namely Semi-AE and Semi-1DCAE. From Fig. 11, it can be observed that there is a significant amount of

**Table 2 Spectral line interval based on different FlowSpectrum models.**

|  | Nonvpn_chat | Nonvpn_email | Nonvpn_file | Nonvpn_p2p | Nonvpn_streaming | Nonvpn_voip |
|---|---|---|---|---|---|---|
| Semi-2DCAE | [8.620, 13.953] | [19.282, 38.120] | [−12.341, −7.198] | [2.165, 3.442] | [−2.782, −0.775] | [−24.549, −15.813] |
| Semi-1DCAE | [−27.591, −4.223] | [−32.851, −18.023] | [0.686, 5.242] | [−4.785, 0.877] | [−2.162, 2.743] | [1.594, 10.248] |
| Semi-AE | [0.347, 7.457] | [9.394, 22.308] | [−73.894, −33.535] | [14.079, 88.715] | [−10.455, 0.109] | [−30.963, −8.650] |
|  | vpn_chat | vpn_email | vpn_file | vpn_p2p | vpn_streaming | vpn_voip |
| Semi-2DCAE | [−9.276, −4.831] | [−1.912, −0.365] | [1.460,5.195] | [−3.471, −1.944] | [−0.356, 1.024] | [14.299, 19.061] |
| Semi-1DCAE | [−0.514, 1.765] | [−1.809, 0.353] | [−4.103, −0.229] | [−2.077, −0.448] | [−1.870, −0.292] | [3.219, 5.612] |
| Semi-AE | [13.173, 34.872] | [−9.157, −1.847] | [−50.462, −11.081] | [−6.193, −0.364] | [−22.469, −6.964] | [0.7946, 7.900] |

overlap between different types of FlowSpectrum. For example, in Fig. 11A, there are file, streaming, and VoIP traffic types distributed within the interval (−40, 0). In Fig. 11C, the chat, email, file, P2P, streaming, and VoIP traffic types are all distributed within the range of (−2, 2). Indeed, it is evident that there is a substantial overlap in the feature representation spectra of encrypted flows generated by the benchmark models, Semi-AE and Semi-1DCAE. The specific numerical values of the FlowSpectrum interval generated by the Semi-2DCAE, Semi-AE, and Semi-1DCAE models are presented in Table 2. From the Table 2, it is evident that only the Semi-2DCAE model is capable of distinguishing different types of traffic into distinct ranges, while the other models fail to fully separate them. The Semi-2DCAE model demonstrates satisfactory results in representing encrypted flow features, as the FlowSpectrum of different types correspond to different intervals in the one-dimensional coordinate system. This confirms the effectiveness of our Semi-2DCAE model in capturing, decomposing, and reducing the dimensional representation of the original spatial structural features of encrypted network flows. In contrast, the Semi-AE and Semi-1DCAE benchmark models exhibit limitations in this regard.

To further analyze the effectiveness of FlowSpectrum in representing the low-dimensional features of encrypted network flows, we employed the Semi-2DCAE, Semi-AE, and Semi-1DCAE models to visualize the ISCX-VPN2016 dataset. As described in 'Mapper', the information contained in the raw network flow data can be divided into representable feature $\vec{T}$ and non-representable feature $\vec{F}$. Based on our Semi-2DCAE model, the two-dimensional visualization of the ISCX-VPN2016 dataset is presented in Fig. 12, where the vertical and horizontal axes represent $\vec{T}$ and $\vec{F}$, respectively. From Fig. 12, it is evident that data points of different types are separable along the vertical axis. However, the distribution of data points along the horizontal axis appears to be chaotic. This provides convincing evidence that the representable feature $\vec{T}$ and the non-representable feature $\vec{F}$ have been successfully separated, thus demonstrating the effectiveness of our Semi-2DCAE model. For comparison, we have reproduced the visualization results of encrypted flows based on the Semi-AE and Semi-1DCAE models. The visualization results are shown in Figs. 13 and 14. From these figures, it can be observed that both of these models exhibit poor separation of representable features in encrypted flows.

In summary, after comparing the FlowSpectrum, spectrum interval values, and two-dimensional data visualization of the Semi-2DCAE, Semi-AE, and Semi-1DCAE models, it

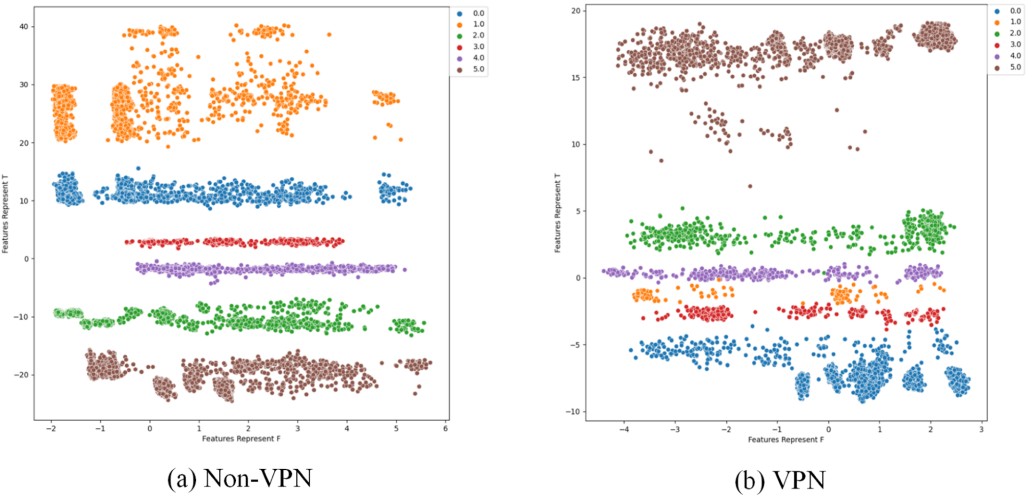

(a) Non-VPN          (b) VPN

**Figure 12 Two-dimensional visualization of traffic features based on Semi-2DCAE model.** (A) Non-VPN. (B) VPN. The labels "0.0", "1.0", "2.0", "3.0", "4.0" and "5.0" respectively represent chat, email, file, p2p, streaming, and voip flow types.

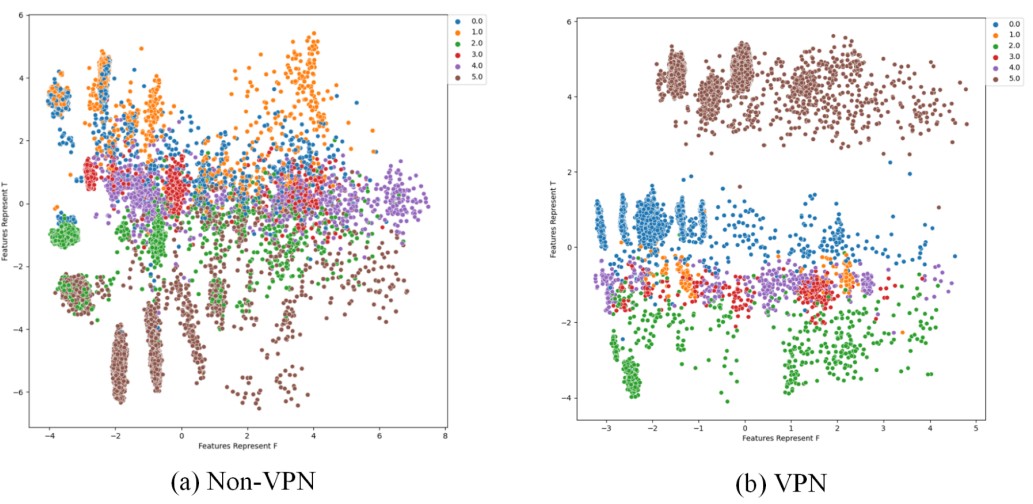

(a) Non-VPN          (b) VPN

**Figure 13 Two-dimensional visualization of traffic features based on Semi-1DCAE model.** (A) Non-VPN. (B) VPN. The labels "0.0", "1.0", "2.0", "3.0", "4.0", and "5.0" respectively represent chat, email, file, p2p, streaming, and voip flow types.

can be determined that the FlowSpectrum generated by the proposed Semi-2DCAE model provides the best representation of the features in encrypted flows.

## Comparison of encrypted traffic classification

As shown in 'Comparison of feature representation of encrypted traffic' above, the FlowSpectrum characterising encrypted flows based on different FlowSpectrum models. In this section, we will classify the encryption using the FlowSpectrum and do a comparative

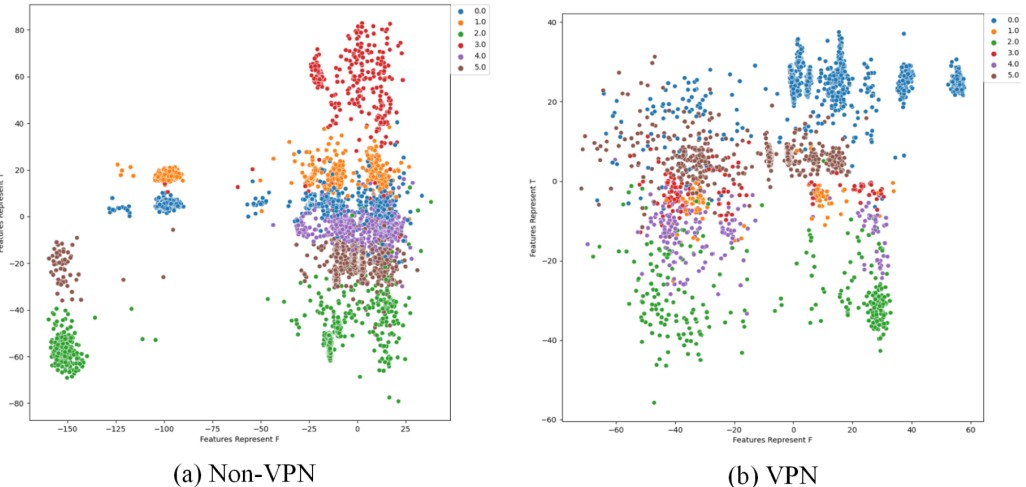

(a) Non-VPN                                          (b) VPN

**Figure 14   Two-dimensional visualization of traffic features based on Semi-AE model.** (A) Non-VPN. (B) VPN. The labels "0.0", "1.0", "2.0", "3.0", "4.0", and "5.0", respectively represent chat, email, file, p2p, streaming, and voip flow types.

**Table 3   Spectral line interval based on different FlowSpectrum models.**

| | NonVpn | | | | Vpn | | | |
|---|---|---|---|---|---|---|---|---|
| | Accuracy | Precision | Recall | F1-score | Accuracy | Precision | Recall | F1-score |
| SVM | 79 ± 0.75% | 84 ± 0.45% | 83 ± 0.71% | 83 ± 0.21% | 95 ± 0.21% | 95 ± 0.44% | 88 ± 0.74% | 91 ± 0.80% |
| 1D-CNN | 72 ± 0.73% | 74 ± 0.28% | 76 ± 0.18% | 74 ± 0.68% | 85 ± 0.10% | 75 ± 0.38% | 60 ± 0.70% | 65 ± 0.16% |
| CNN+RNN | 90 ± 0.15% | 90 ± 0.57% | 90 ± 0.73% | 90 ± 0.47% | 96 ± 0.71% | 97 ± 0.30% | 96 ± 0.21% | 96 ± 0.51% |
| Semi-AE | 81 ± 0.85% | 85 ± 0.19% | 81 ± 0.86% | 79 ± 0.92% | 95 ± 0.32% | 95 ± 0.62% | 95 ± 0.33% | 95 ± 0.25% |
| Semi-1DCAE | 80 + 0.04% | 80 ± 0.51% | 80 + 0.04% | 79 ± 0.31% | 83 ± 0.96% | 84 ± 0.23% | 84 ± 0.15% | 82 ± 0.87% |
| Semi-2DCAE | 99 ± 0.17% | 99 ± 0.23% | 99 ± 0.17% | 99 ± 0.17% | 98 ± 0.25% | 98 ± 0.05% | 97 ± 0.30% | 97 ± 0.21% |

analysis with the other five benchmark (Semi-AE, Semi-1DCAE, 1DCNN, SVM and CNN+RNN) models.

As shown in Table 3, it is the average of the results after 10 classification of Non-VPN encrypted flows and VPN encrypted flows. Among them, when classifying non-VPN encrypted flows, the classification result based on our model reaches 99.2%, which is the best, and the classification result based on 1D-CNN is 72.7%, which is the worst. In the classification results of VPN encrypted flows, our model's classification result reached 98.3%, which is also the best. The classification result based on Semi-1DCAE model is 84.0%, which is the worst. Table 3 shows the effectiveness of our FlowSpectrum model. Our classification scheme is better than ML and DL based classification schemes on the whole.

The above presents a comparison of each model's overall performance. We now analyse the classification performance of the six schemes for different flow types on the two types of encryption. Therefore, we illustrate the confusion matrices of these methods on the two types of encrypted streams in Figs. 15 and 16. These results also confirm the superiority

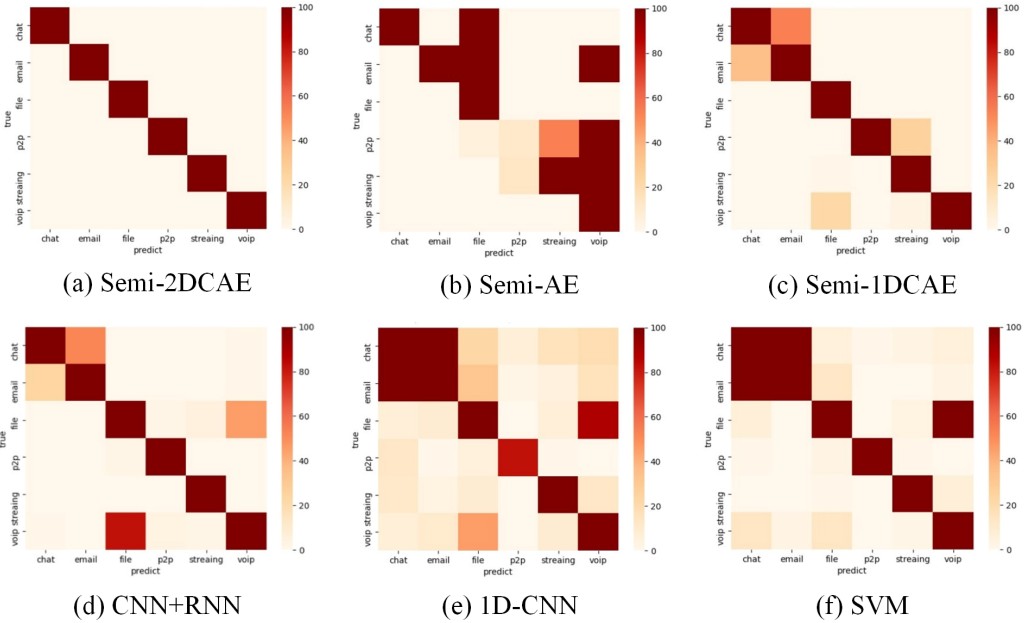

**Figure 15  Confusion matrix of Non-VPN encrypted traffic based on different classification models.** (A) Semi-2DCAE, (B) Semi-AE, (C) Semi-1DCAE, (D) CNN+RNN, (E) 1D-CNN, (F) SVM. The abscissa represents the prediction and the ordinate indicates the true label. The darker color of the elements on the main diagonal, the better performance.

of encrypted traffic classification based on Semi-2DCAE. In Figs. 15A and 16B, we can see that the main diagonal is the darkest in colour. Moreover, we find that in Fig. 16A, there is a shadow at the intersection of P2P and Chat. This is that part of P2P data is predicted to be Chat data. We think that the reason for this situation is that the features in P2P and the features in the Chat flow partially overlap, resulting in classification errors.

## CONCLUSION

As an emerging technology for network flow representation, FlowSpectrum is a method of mapping high-dimensional features of network space flows into low-dimensional representations. Using FlowSpectrum, various network flow analyses such as network flow classification can be performed.

In the early days, network flow classification relied mainly on port numbers or DPI technology. However, with the development of networks and the increasingly pressing issue of network security, these methods have encountered significant challenges, as ports are often abused and encryption is increasingly used in communications. In recent years, the rise of machine learning and deep learning has provided more solutions for network flow classification. In ML-based classification, network flow features need to be filtered manually, and then ML models (SVM, DT, KNN) are used as classifiers for classification. In the classification based on DL models (CNN, RNN), the end-to-end mode is used to replace manual feature selection. However, while ML and DL have greatly improved the accuracy and efficiency of network classification, little research seems to have focused

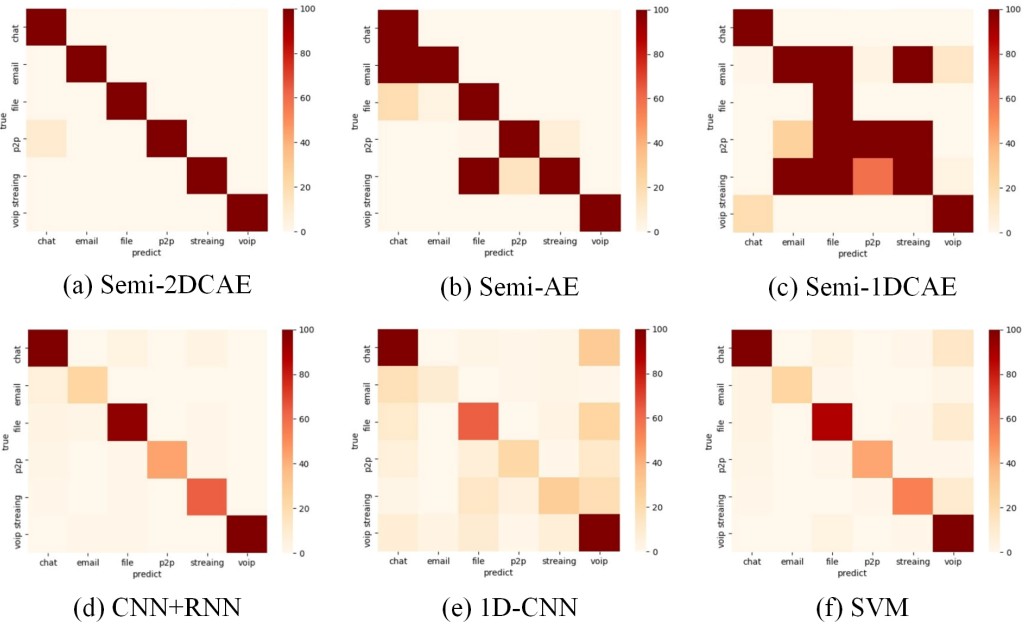

**Figure 16  Confusion matrix of VPN encrypted traffic based on different classification models.** (A) Semi-2DCAE, (B) Semi-AE, (C) Semi-1DCAE, (D) CNN+RNN, (E) 1D-CNN, (F) SVM. The abscissa represents the prediction and the ordinate indicates the true label. The darker color of the elements on the main diagonal, the better performance.

on the characterization of features by these methods, or on describing features as well as providing interpretability, especially in end-to-end DL. As a new scheme of network flow analysis, the FlowSpectrum theory solves the above problems well. However, existing FlowSpectrum models still face many challenges such as poor generalization ability, weak model feature extraction ability, and low classification accuracy for encrypted traffic.

In this article, we propose a Semi-supervision 2D-CNN AutoEncoder (Semi-2DCAE) method to address these challenges. First of all, in order to make the FlowSpectrum more characterize characteristics of encrypted flows, we use the method of full-layer information extraction of data packets and generate IDX files that can be processed by 2D-CNN during data processing. After that, we use the Semi-2DCNN model to extract and reduce the spatial structure features of the data, and then establish a standard one-dimensional coordinate system to characterize the features extracted from the model as spectral lines, which we call FlowSpectrum. Finally, the FlowSpectrum is utilized to classify the test data. Experimental results demonstrate that compared to existing schemes, the FlowSpectrum generated based on Semi-2DCNN successfully characterizes different types of encrypted flows and achieves higher accuracy in their classification.

In the future, our work will be focused on three main areas.

(i) FlowSpectrum characterisation

The key to the quality of FlowSpectrum for characterising network flows is in the extraction and representation of representable information. The flows in the network space are complex and varied, with thousands of features, and finding a more appropriate

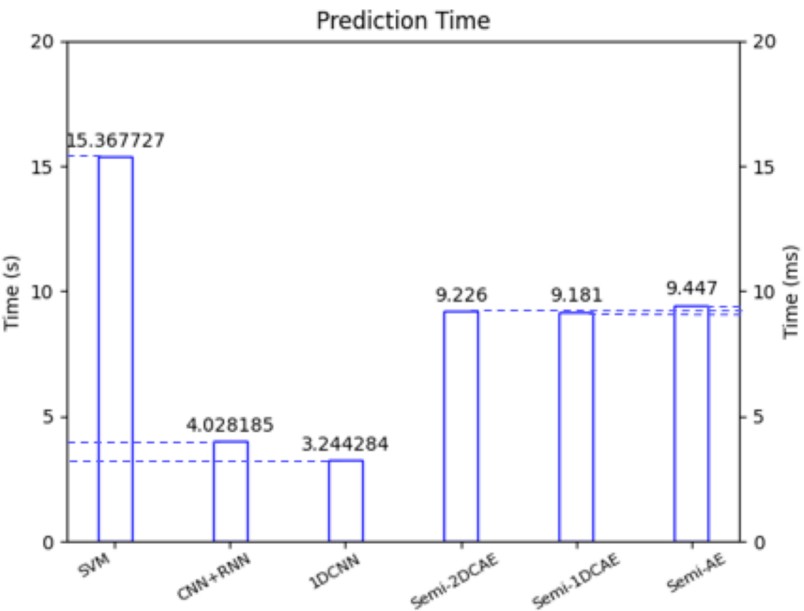

**Figure 17** **Classification delay of different models.** Note that here the left vertical axis is in seconds (S) and the right vertical axis is in milliseconds (ms).

mapping approach may allow better and more characterisation of the network flow features. We argue that the mapping approach is not just a mapping of time and space, as in *Fu et al. (2022)* where the network flows are extracted through Fourier variation to extract the frequency domain features of the flows, and in *Agrawal & Tapaswi (2020)* where the researchers use Fast Hartley Transform (FHT) to extract the frequency domain features of the flows. As mentioned above, the mapping of network flows into one-dimensional coordinates in our model is a series of transformations to obtain one-dimensional features from the spatial features of the flows. In the future, we will explore multi-dimensional spectral line features including two-dimensional and three-dimensional.

(ii) Online analysis

In the future, we will utilize FlowSpectrum for the online classification of network flows. One of the key challenges in online network flow analysis is classification latency. In reference (*Xie, Li & Jiang, 2021*), the latest online analysis approach was employed, and according to the article's data, the online classification latency was approximately 2 ms. In reference (*Fu et al., 2022*), researchers conducted online attack detection on network flows using an online packet-capturing tool. Figure 17 illustrates the latency of our Semi-2DCAE model and five other benchmark models for encrypted traffic classification. Specifically, the FlowSpectrum classification latency based on the Semi-2DCAE model is 9.226 ms, which provides possibilities for our future work on online traffic analysis. (iii) Unknown network flow classification

At present, we can only classify network flow types that are known to us. However, we are unable to identify flow types that are unknown. Therefore, our future research will primarily focus on classifying unknown network flows.

### Funding
The authors received no funding for this work.

### Competing Interests
The authors declare there are no competing interests.

### Author Contributions
- Jun Cui conceived and designed the experiments, performed the experiments, analyzed the data, performed the computation work, prepared figures and/or tables, authored or reviewed drafts of the article, and approved the final draft.
- Longkun Bai conceived and designed the experiments, performed the experiments, analyzed the data, performed the computation work, prepared figures and/or tables, authored or reviewed drafts of the article, and approved the final draft.
- Guangxu Li performed the experiments, authored or reviewed drafts of the article, and approved the final draft.
- Zhigui Lin conceived and designed the experiments, authored or reviewed drafts of the article, and approved the final draft.
- Penggao Zeng performed the computation work, authored or reviewed drafts of the article, and approved the final draft.

### Data Availability
The code is available at Zenodo: blank. (2023). Semi-2DCAE [Data set]. Zenodo. https://doi.org/10.5281/zenodo.8154325.

The data is available at VPN-nonVPN dataset (ISCXVPN2016): https://www.unb.ca/cic/datasets/vpn.html.

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
