# Peer review of "Semi-2DCAE: a semi-supervision 2D-CNN AutoEncoder model for feature representation and classification of encrypted traffic"

_PeerJ Computer Science, doi:10.7717/peerj-cs.1635_

## Round 0.1 · original submission · Major Revisions

Please ensure the second and third reviewers' comments are addressed.
Thank you
C.S.

Reviewer 1 ·

Basic reporting

The contribution of the paper is good thus it is sufficient to be considered for publication

Experimental design

No comment

Validity of the findings

No comment

Additional comments

No comment

Reviewer 2 ·

Basic reporting

This paper requires careful proofreading. The following issues need to be addressed to pass the review.

Lines 81-86 discuss prior work on flowspectrum, and lines 87 onwards talk about the current work. It is not mentioned how the current work is different from the prior work on DL-based flowspectrum. What are the challenges with the prior work that Semi-2DCAE is solving?
This has to be addressed in the Related Work section also. It should clearly state what the limitations of the existing DL flowspectrum methods are the current work addressing.

Line 185: What is certain accuracy?

Line 194: citation seems to be wrong. Work referred to is Guo et al but the citation mentions Yang et al.

Line 269: xi is the instance flow point value instead of fi.

Figure 4: The description in Lines 257-264 is not consistent with the Figure. Writeup says multiple Session files but only one is shown in the Figure. The packet Pick part of the figure shows only one file with 0 fill in the pipeline but the next part, IDX Generate shows multiple images.

Page 7 para 1: Should the dimensionality of the instance output set be k instead of m?

Line 272: It should be "then" instead of "hen".

page 8 line 1: what is "mat hitt"?

Figures 7 and 8: What is result accuracy? How is the validation set selected?

Table 1: "Everage" should be "Average"

Figure 9 should state that statistical features are used in the experiment.

Figure 10: Both subfigures have the same caption. What is the difference between the two setups?

Line 359: What are the two types of flow spectrum? Is it the vpn and the non-vpn? This should be clarified in the write-up and in the figures.

Line 362: What are the other two models?

Figure 11: Subfigures (a) and (b) have the same caption and subfigures (c) and (d) also have the same caption. Each figure should have distinct captions.

Line 370: "We" should be "we".

Line 371: what are the three types of flow spectrum models? It should be clearly stated that two are baselines and one is your work.

Figures 12, 13, 14: Subfigures should have distinct captions. Are they representing results for vpn and non-vpn traffic?

Figure 16: captions of subfigures look like variable names used in programs. Please use better captions.

Line 410: Fix the sentence.

Experimental design

No comment.

Validity of the findings

What are the internal and external threats to the validity of this work? This discussion should be added to the paper.

The artifact does not have enough documentation. For example, two test and train scripts are provided. It is not mentioned in the readme file what these files are for. What is the test script's objective? Does the results map to any figures in the paper? Add details to the readme file.

Reviewer 3 ·

Basic reporting

-The grammar and writing in this paper need to be polished.
-The story telling in the paper cannot convince the readers.
- The related work part does not make a significant contribution. There is no in-depth details of their advantages and disadvantages. Moreover, the authors neglect to demonstrate how this section can inspire them to develop the contributions they have already made.

Experimental design

- The results in Fig.17 are not relieable. So, time complexity must be analyzed for supporting the method's results.

Validity of the findings

- Contributions are not clear and they are not solid.

Additional comments

I cannot see any theoretical development, or remarkably valuable finding. The authors must demonstrate the reliability of their work. Although it might lead to some improvements, it cannot be regarded as a significant contribution or new finding. Scientific papers should provide advanced in the state of the art, and I think that this paper does not reach to that level.

---

## Round 0.2 · accepted · Accept

The paper at the current revision is now suitable for PeerJ publication. Thank you for your submission.

Reviewer 2 ·

Basic reporting

no comment

Experimental design

no comment

Validity of the findings

no comment

Additional comments

All the comments that I provided in my previous review were adequately addressed.

Reviewer 3 ·

Basic reporting

'no comment'

Experimental design

There is more clarity and detail in this revised version.

Validity of the findings

The fidings are clear.

Additional comments

The comments raised by the previous round reviewers have been addressed.